# Lung Ultrasound in the Screening of Pulmonary Interstitial Involvement Secondary to Systemic Connective Tissue Disease: A Prospective Pilot Study Involving 180 Patients

**DOI:** 10.3390/jcm10184114

**Published:** 2021-09-12

**Authors:** Natalia Buda, Anna Wojteczek, Anna Masiak, Maciej Piskunowicz, Wojciech Batko, Zbigniew Zdrojewski

**Affiliations:** 1Department of Internal Medicine, Connective Tissue Diseases and Geriatrics, Medical University of Gdansk, 80-214 Gdansk, Poland; natabud@gumed.edu.pl (N.B.); anna.wojteczek@gumed.edu.pl (A.W.); zzdroj@gumed.edu.pl (Z.Z.); 2Radiology Department, Medical University of Gdansk, 80-214 Gdansk, Poland; maciej.piskunowicz@gumed.edu.pl; 3First Department of Cardiology, Medical University of Gdansk, 80-214 Gdansk, Poland; wojciech.batko@gumed.edu.pl

**Keywords:** LUS, HRCT, ILD, pulmonary fibrosis, chest sonography

## Abstract

Objectives: The aim of the study was the assessment of lung ultrasound (LUS) as a screening of pulmonary interstitial involvement secondary to systemic connective tissue diseases. Methods: A prospective study was conducted on the study group comprising 180 patients diagnosed with different systemic connective tissue diseases. Each patient underwent lung ultrasound (LUS), high-resolution chest computed tomography (HRCT), and echocardiography (ECHO). Each imaging examination was blinded and performed by an independent operator. LUS was conducted with convex and linear transducers. Results: The sensitivity and specificity of LUS as compared to HRCT in detecting pulmonary interstitial involvement in the study group were 99.3% and 96.4%, respectively; positive predictive value (PPV) 0.7, negative predictive value (NPV) 3.6. Abnormalities indicating interstitial lung disease (ILD) with fibrosis were most frequently localized bilaterally in the lower fields of the lungs, assessed in the dorsal view. Conclusions: LUS is an efficient imaging modality that can detect pulmonary interstitial involvement in patients with systemic connective tissue disease with a high sensitivity and specificity. Further prospective studies conducted on a larger population are deemed necessary.

## 1. Introduction

Lower respiratory tract involvement is a frequent problem occurring in systemic connective tissue diseases [1]. Interstitial lung disease (ILD) is often detected in this patient group, with the incidence depending on the rheumatic disease diagnosis [2]. Simultaneously, it has been demonstrated that about 15% of patients with ILD develop this condition secondary to systemic connective tissue disease [3]. The average period between the first symptoms and diagnosis is about 1–2 years [3]. High resolution chest computed tomography (HRCT) is the gold standard in ILD diagnosis. However, owing to the limited access to HRCT, ionizing radiation, the cost of the examination and the need to perform it mostly in radiological units, it is justifiable to search for other lung imaging methods. The results of studies devoted to the application of lung ultrasonography (LUS) within the last decade have been very promising [4,5,6]. Consequently, we conducted a blind prospective study, aimed at evidencing the effectiveness of LUS in detecting ILD in patients with different systemic connective tissue diseases.

## 2. Materials and Methods

### 2.1. Study Group

Patients qualified for the study were diagnosed with different systemic connective tissue disease. All of them met the classification criteria of the particular types of diseases and had a stable form of the disease for at least 2 years. Additional inclusion criteria included age over 18 years and written consent for participation in the study.

Patients who did not consent for participation in the study, as well as pregnant patients and those who had a CT scan performed on any body part within the previous year were excluded. Moreover, patients with a diagnosis of clinically significant left ventricular failure or valvular heart defect (serious mitral or aortic valve disease) detected in echocardiography (ECHO) were also excluded from the final analysis of the results.

### 2.2. Ethics Committee

The research project was granted consent by the independent Ethics Committee, number NKBBN/483/2019.

### 2.3. Imaging Examinations: Methodology

#### 2.3.1. LUS Protocol

LUS examinations were performed with an ultrasound device (Philips, 2013, NY, USA), equipped with two transducers. Convex transducers with a frequency of 2–6 MHz and linear transducers with a frequency of 4–12 MHz were employed. LUS examinations were performed in each patient in a sitting and supine position. The transducer was placed over each intercostal space available during the examination and over supraclavicular fossa. The chest was divided into 16 zones (Figure 1). The analyzed findings included: abnormalities within the pleural line (irregular, coarse, fragmented and blurred pleural line), vertical artifacts (B-lines, Z-lines, C-lines) and consolidations (<5 mm and >5 mm). The pathologies detected in LUS were recorded on a dedicated examination form. Then data concerning LUS findings were statistically analyzed. The coexistence of bilateral B-line artifacts (single or multiple) and pleural line disorders (irregular pleural line and coarse or fragmented or blurred) were assumed as “positive LUS results” for pulmonary fibrosis.

#### 2.3.2. HRCT Protocol

Chest CT scans were obtained following the standard protocol employed in our Radiology Department for high-resolution lung CT (HRCT). Images were acquired with a 128-detector row scanner SOMATOM Definition Flash (Siemens, Forchheim, Germany) in the craniocaudal direction during a single breath-hold with collimation 128 × 0.6 mm, rotation time 0.5 s, matrix 512 × 512 mm^2^, and 0.6 mm slice thickness (gapless). Lung HRCT image analysis was performed using dedicated software (Syngo.via, Siemens, Germany) and an application (CT-Chest; Syngo.via) with standard lung window settings (width, −50 HU; level, 1500 HU) and mediastinal window settings (width, 350 HU; level, 50 HU). CT scans were reviewed by a radiologist (M.P.) with 19 years of experience. The following data were reported according to the specified lung regions: the presence of septal thickening; bronchiectasis; honeycombing; ground-glass opacity; pleural thickening; tree-in-bud sign; consolidation; calcification, and pulmonary emphysema. ILD with pulmonary fibrosis or “positive HRCT”, was considered when: bilateral reticular pattern and/or honeycomb lesions and/or bronchiectasis were found. Moreover, radiological diagnosis other than ILD with fibrosis were ruled out (for example pneumonia, lymphangitis carcinomatosa or lung cancer).

#### 2.3.3. ECHO Protocol

A complete transthoracic echocardiogram was performed using a GE VIVID E9 ultrasound system (GE Ultrasound, Horten, Norway) equipped with a phased-array transducer (M5S). Standard echocardiographic parameters were obtained according to the principles described in the ASE/EACVI recommendations [7]. Left ventricular ejection fraction (LVEF) was measured using the apical biplane Simpson’s method. LV dysfunction was defined as LVEF < 52% for men and LVEF < 54% for women, consistent with the current recommendations [7]. Left ventricular diastolic function and the severity of valvular regurgitation (AR, MR, TR) and aortic stenosis (AS) were assessed using an integrated method consistent with the established practice guidelines [8,9,10]. The systolic pulmonary artery pressure (SPAP) was estimated using TR peak velocity and right atrial pressure, which was estimated by the inferior vena cava diameter in a long-axis subxiphoid view and its response to inspiration. All echocardiograms were stored digitally, and further offline analysis was performed using an EchoPAC workstation (v201, GE Healthcare, Horten, Norway).

#### 2.3.4. Statistical Analysis

Statistical analyses were performed with IBM SPSS Statistics 25.0 software (Armonk, NY, USA). This software was employed to analyze the frequency of pulmonary lesions detected in HRCT and LUS. To determine the relationship between abnormalities detected in HRCT and in LUS, a chi-square test was employed or Fisher’s exact test (when the expected number was smaller than 5) with the phi Yule coefficient as a measure of the power of correlation. Statistical significance was assumed with α = 0.05.

## 3. Results

### 3.1. Study Group

Out of 257 patients diagnosed with systemic connective tissue disease, 77 individuals were excluded due to CT scan of any body part performed within the previous year (chest, abdomen, head, neck), which was one of the exclusion criteria. In total, 180 patients participated in the study, with a further four patients being excluded due to their ECHO results, indicating identified de novo left ventricular failure (two patients with LVEF < 50%, and one person with grade III left ventricular diastolic dysfunction and one person with LVEF < 50% and diastolic dysfunction) with coexisting severe valvular heart disease.

Diagnosis of rheumatic diseases in the study group was determined consistent with the current American College of Rheumatology (ACR) and European League Against Rheumatism (EULAR) diagnostic guidelines. Table 1 demonstrates in detail the characteristics of the study group. There were only seven active smokers among the patients studied, which did not affect the study results.

### 3.2. Results of Imaging Examinations

#### 3.2.1. ECHO Results

Table 2 contains the analysis of echocardiographic findings in the study group. The analysis revealed that 12% (*n* = 22) of patients demonstrated the IVC width expiration/inspiration < 50%, and 6.3% (*n* = 11) IVC > 21 mm. LVEF < 50% occurred in three patients and in the same number of patients right ventricular (RV) systolic function < 9.5 cm/s and a serious defect of the aortic valve was detected. In 3.4% (*n* = 6) of patients RV systolic function TAPSE < 17mm was revealed, and in 2.3% (*n* = 4) a serious defect of the mitral valve. The lesions in ECHO overlapped in individual patients, for example: among patients with heart failure there was serious mitral valve defect in three patients (MR ≥ moderate or stenosis) and severe defect in both the aortic and mitral valve in one patient.

#### 3.2.2. LUS and HRCT Results

The LUS examination took an average of 5–7 min. On the other hand, the description of the test result for medical records took an average of 3 min. Below we present sets of lesions occurring in imaging tests in LUS and HRCT.

Table 3 demonstrates bilateral findings involving the pleural line detected in LUS and peripheral changes observed in HRCT.

An irregular pleural line was revealed in 19.3% (*n* = 34) of patients, coarse in 16.5% (*n* = 29), and fragmented in 10.25% (*n* = 18). Blurred pleural line was found in seven patients and thickened in one person. Single artifacts were visualized in 15.9% (*n* = 28) of participants, and multiple artifacts in 9.7% (*n* = 17). The white lung sign was detected in 2.3% (*n* = 4) of patients. Am-line artifacts were revealed in one patient, and consolidations < 5 mm in 7 patients, whereas > 5 mm in two patients.

In HRCT, reticular pattern was revealed in 16.5% (*n* = 29) of patients, and small-nodular pattern only in three patients (1.7%). Interlobular septal thickening was found in 2.8% (*n* = 5) of patients, and bronchiectasis, changed by inflammation, in 14.2% (*n* = 25). Cysts were visible in 20 patients (11.4%) and honeycombing in seven (4%). Ground-glass opacities were visualized in 4.5% (*n* = 8) of patients.

Considering LUS findings, the most frequent abnormalities involved irregular and coarse pleural lines, as well as multiple B-line artifacts, and were revealed in more than 15% of participants. The most frequent HRCT findings were reticular pattern and bronchiectasis, changed by inflammation, as well as cysts.

To determine the correlation between LUS and HRCT results, independence analysis was performed with a chi-square test. The analysis revealed a significant correlation between the variables, χ^2^(1) = 161.37; *p* < 0.001; *φ* = 0.96. Among 28 cases positively diagnosed in HRCT, 27 were also positively classified based on LUS findings (Table 4). In the case of one person, interstitial pneumonia was diagnosed based on HRCT, but not on LUS, and in one case such diagnosis was made based on LUS, which was not confirmed by HRCT. This result indicates that the classification correctness for these methods amounts to 98.9%, whereas the results are concordant in 96.4%. LUS findings were compared with those of HRCT statistically, and a sensitivity and specificity of LUS for detecting pulmonary interstitial involvement in the study group were calculated, the results being 99.3% and 96.4%, respectively; PPV 0.7, NPV 3.6.

Positive HRCT results were obtained in 28 patients, and for this group additional calculations were performed, including the localization of abnormalities.

#### 3.2.3. Correlation of Findings in LUS and HRCT in the Group with Positive HRCT Results

Next, the analysis with Fisher’s exact test was conducted, with the phi coefficient, indicating the power of variables correlation. Analyses were performed separately for each localization.

Table 5 demonstrates the relationships between the frequency of pathological findings in HRCT and LUS. The analysis revealed strong correlations between the reticular pattern in HRCT and the presence of the irregular, coarse and fragmented pleural line, as well as single and multiple B-line artifacts visualized in LUS. Interlobular septal thickening was most strongly correlated with the white lung. Bronchiectasis, changed by inflammation, strongly correlated with the irregular, coarse and fragmented pleural line. Honeycombing strongly correlated with the presence of consolidations <5 mm, B-line artifacts forming the white lung and blurred pleural line. The remaining correlations were either moderate or weak. All discussed correlations were positive: the more often specific findings were found in HRCT, the more often they were detected in LUS. The analysis showed that a positive ultrasound result was recorded in the majority of cases when the following symptoms were present: irregular pleural line, coarse, fragmented pleural line and single and multiple B artefacts. The relative risk of interstitial disease ranged from 4.82 for irregular pleural line to 15.72 for multiple B artifacts (Table 6).

#### 3.2.4. Number of Particular Abnormalities Detected in LUS in Specific Localizations

Finally, the analysis for the number of abnormalities observed in LUS in specific localizations was performed (Table 7). The analysis revealed that the majority of abnormalities occurred in two localizations: bilaterally in the lower field of the lungs, dorsally and paraspinally. As regards irregular, coarse, fragmented and blurred pleural lines as well as single B-line artifacts, multiple B-line artifacts, white lung and consolidations, these abnormalities were detected with a similar frequency in the lower field of both lungs, dorsally, between the scapular line and posterior axillary line. In the remaining localizations, abnormalities observed in LUS were sporadic.

## 4. Discussion

The risk of developing ILD by patients with systemic connective tissue disease is higher than in other populations [2]. The development of ILD and its progression result in pulmonary hypertension and a worse quality of life as well as a shorter life span due to respiratory failure [11]. Consequently, rheumatology patients require careful diagnostics and monitoring focused on ILD. ILD, secondary to systemic connective tissue disease, is progressive, but its progression and tendency towards reversibility following the administered treatment is different than in the case of idiopathic pulmonary fibrosis [12]. Diagnosing ILD at its early stage as well as monitoring its course are crucial for both treatment and prognosis [11].

Most available studies concerning the application of LUS involve a defined group of patients diagnosed with ILD secondary to systemic connective tissue disease [6,13]. Despite single reports on the use of LUS in the detection of ILD, the method still requires studies in larger groups of patients and validation [14].

Nevertheless, data derived from meta-analyses indicate that LUS is of significant diagnostic value in clinical practice and is characterized by a high accuracy for detecting pulmonary fibrosis [15,16].

We decided to conduct a prospective study with blinded results. LUS, HRCT and ECHO examinations were performed by three independent operators and input into 3 independent databases. None of the persons conducting a particular imaging examination knew the results of the remaining imaging methods. Patients were qualified for the study by specialists in rheumatology. The major advantage of the study protocol was the performance of all three imaging examinations for each patient on average within 2 h. LUS findings were compared with those of HRCT statistically, and a sensitivity and specificity of LUS for detecting pulmonary interstitial involvement in the study group were calculated, the results being 99.3% and 96.4%, respectively; PPV 0.7, NPV 3.6. Moreover, it was revealed that the most frequent localization of abnormalities indicating ILD with fibrosis occurred bilaterally in the lower field of the lungs, assessed in a dorsal view.

B-lines are the most frequent LUS finding in the course of ILD [17]. They are excellent markers of pulmonary interstitial involvement, irrespective of the etiology. It should be stressed that B-line artifacts are also found, among others, in patients with cardiogenic pulmonary edema, interstitial pneumonia (e.g., due to viral infection), acute respiratory distress syndrome (ARDS) and diffuse alveolar hemorrhage [18,19]. It should also be noted that B-line artifacts in patients diagnosed with ILD occur both in actively developing lesions (such as ground-glass finding in HRCT) and already existing ones (e.g., honeycombing) [20].

Some studies focused on the assessment of the severity of pulmonary fibrosis based on the number of B-line artifacts [21,22]. Counting B-line artifacts allowed researchers to present a mathematical correlation with the results of HRCT, with the application of the Warick score [4,23]. It was evidenced that the more B-line artifacts were found in each patient, the more severe the pulmonary involvement was. This, in turn, was often associated with the worsened gas exchange observed in pulmonary function tests [24].

Recent publications emphasize the need for observation of abnormalities within the pleural line that constitute the source of vertical artifacts [25,26]. To do that, special attention is given to the application of a linear transducer during the examination [27]. Abnormalities within the pleural line include irregular, coarse, fragmented, blurred and thickened pleural line [28,29]. These abnormalities can be detected with a linear transducer that has an additional diagnostic value essential in differential diagnosis of the etiology of pulmonary interstitial involvement. In our study, the relationships between the frequency of pathological findings in HRCT and LUS were revealed. The analysis revealed strong correlations between the reticular pattern in HRCT and the presence of the irregular, coarse, and fragmented pleural line, as well as single and multiple B-line artifacts visualized in LUS. Interlobular septal thickening was most strongly correlated with the white lung. Bronchiectasis, changed by inflammation, strongly correlated with the irregular, coarse and fragmented pleural line. Honeycombing strongly correlated with the presence of consolidations <5 mm, B-line artifacts forming the white lung and blurred pleural line. Referring to the data from the 2019 meta-analysis, which analyzed 11 studies involving 487 CTD patients, the results of the current work are consistent and similar. The total sensitivity and specificity of the LUS were 0.859 (95% confidence interval (CI) 0.812–0.898) and 0.839 (95% CI 0.782–0.886), respectively. It should be noted, however, that most of the studies present the incidence and quantitative or semi-quantitative determination of presence and the number of B lines. Presentation of the ultrasound model of interstitial lung disease is still not precise [15,16]

Our study has some limitations. One of them with a relatively small number of patients. Another limitation of the method is the assessment only on the basis of artifacts and changes occurring superficially in the area of the pleural line and just below it. HRCT, when compared to LUS, is a method that perfectly shows lesions in the course of ILD cross the full lung profile. LUS is a method that depends on the test technique and the experience of the person performing the test. The technique of performing the HRCT examination is well developed and standardized in every computer tomography laboratory, but also here the experience of the radiologist in developing the examination result is important. Moreover, ILD secondary to systemic connective tissue disease may have various histopathological patterns. Usual interstitial pneumonia (UIP) and non-specific interstitial pneumonia (NSIP) are most common. Owing to the lack of histopathologic examination results, differential analysis assessing LUS findings in relation to the histopathological type was not performed.

## 5. Conclusions

LUS is a fast (the examination takes on average 5–7 min), efficient imaging modality that can detect pulmonary interstitial involvement in patients with systemic connective tissue disease with a high sensitivity and specificity. Further prospective studies conducted on a larger population are deemed necessary.

## Figures and Tables

**Figure 1 jcm-10-04114-f001:**
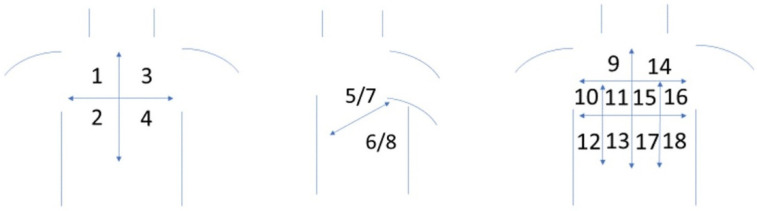
Schematic drawing showing the division of the thorax into pulmonary fields (from 1 to 18), which were assessed equally on lung ultrasound and HRCT (1—right side, upper-anterior field, 2—right side, lower-anterior field, 3—left side, upper-anterior field, 4—left side, lower-anterior field, 5—right side, upper-lateral field, 6—right side, lower-lateral field, 7—left side, upper-lateral field, 8—left side, lower-lateral field, 9—left side, upper-posterior field, 10—left side, middle field, 11—left side, middle-paravertebral field, 12—left side, lower field, 13—left side, lower-paravertebral field, 14—right side, upper-posterior field, 15—right side, middle-paravertebral field, 16—right side, middle field, 17—right side, lower-paravertebral field, 18—left side, lower field).

**Table 1 jcm-10-04114-t001:** Characteristics of the study group.

Parameter	*n* (%)
Gender	
Females	130 (73.9)
Males	46 (26.1)
Rheumatoid arthritis (RA)	22 (12.5)
Eosinophilic granulomatosis with polyangiitis (EGPA)	10 (5.7)
Granulomatosis with Polyangiitis (GPA) and Microscopic Polyangiitis (MPA)	25 (14.2)
Myositis	18 (10.2)
Systemic lupus erythematosus (SLE)	25 (14.2)
Systemic sclerosis (SSC)	30 (17)
Sjögren syndrome	46 (26.1)

**Table 2 jcm-10-04114-t002:** Analysis of echocardiographic findings in the study group (*n* = 176).

	*n*	%
LVEF < 50%	3	1.7
Left ventricle diastolic disfunction, grade III	1	0.57
IVC > 21 mm	11	6.3
IVC width expiration/inspiration <50%	22	12.5
RV systolic function: TAPSE < 17 mm	6	3.4
RV systolic function: s′ < 9.5 cm/s	3	1.7
Serious mitral valve defect (MR ≥ moderate or stenosis)	4	2.3
Serious aortic valve defect (AR ≥ moderate or stenosis)	3	1.7
ILD in CT	28	15.9

**Table 3 jcm-10-04114-t003:** Analysis of pulmonary lesions in LUS and CT. Definition used: Single B lines—less than three in one intercostal space; Multiple B lines—more than three in one intercostal space; White lung—numerous B lines, which merging in a one wide vertical artifact, fill up entire intercostal space.

	*n*	%
LUS		
Pleural line—irregular	34	19.3
Pleural line—coarse	29	16.5
Pleural line—fragmented	18	10.2
Pleural line—blurred	7	4.0
Pleural line—thickened	1	0.6
B-line artifacts—single	28	15.9
B-line artifacts—multiple	17	9.7
B-line artifacts—white lung	4	2.3
Am artifacts	1	0.6
Consolidations < 5 mm	7	4.0
Consolidations > 5 mm	2	1.2
HRCT		
Reticular pattern	29	16.5
Small-nodular pattern	3	1.7
Interlobular septal thickening	5	2.8
Bronchiectasis, changed by inflammation	25	14.2
Cysts	20	11.4
Honeycombing	7	4.0
Ground-glass	8	4.5

**Table 4 jcm-10-04114-t004:** Analysis of diagnosing ILD based on LUS and HRCT.

		Interstitial Disease in LUS			
		No		Yes	
		*n*	%	*n*	%
Interstitial disease in HRCT	no	147	99.3	1	0.7
	yes	1	3.6	27	96.4

**Table 5 jcm-10-04114-t005:** Analysis of the frequency of findings with the coefficient *φ* for the correlations between findings detected in HRCT and LUS. Values marked by bold font indicate strong positive correlations.

LUS	HRCT													
	Reticular Pattern		Small-Nodular Pattern		Interlobular Septal Thickening		Bronchiectasis, Changed by Inflammation		Cysts		Honeycombing		Ground-Glass	
	*n* (%)	*φ*	*n* (%)	*φ*	*n* (%)	*φ*	*n* (%)	*φ*	*n* (%)	*φ*	*n* (%)	*φ*	*n* (%)	*φ*
Irregular pleural line	250 (93.3)	**0.81**	4 (100.0)	0.10	31 (96.9)	0.28	168 (90.3)	**0.64**	94 (83.2)	0.45	33 (97.1)	0.29	37 (90.2)	0.29
	*p* < 0.001		*p* < 0.001		*p* < 0.001		*p* < 0.001		*p* < 0.001		*p* < 0.001		*p* < 0.001	
Coarse pleural line	183 (68.3)	**0.71**	4 (100.0)	0.13	29 (90.6)	0.32	139 (74.7)	**0.65**	68 (60.2)	0.39	30 (88.2)	0.32	34 (82.9)	0.33
	*p* < 0.001		*p* < 0.001		*p* < 0.001		*p* < 0.001		*p* < 0.001		*p* < 0.001		*p* < 0.001	
Fragmented pleural line	91 (34.0)	**0.50**	1 (25.0)	0.04	24 (75.0)	0.39	75 (40.3)	**0.50**	41 (36.3)	0.34	25 (73.5)	0.39	21 (51.2)	0.29
	*p* < 0.001		*p* < 0.001		*p* < 0.001		*p* < 0.001		*p* < 0.001		*p* < 0.001		*p* < 0.001	
Blurred pleural line	34 (12.7)	0.33	0 (0)	−0.004	12 (37.5)	0.34	33 (17.7)	0.38	12 (10.6)	0.17	22 (64.7)	**0.61**	2 (4.9)	0.04
	*p* < 0.001		*p* < 0.001		*p* < 0.001		*p* < 0.001		*p* < 0.001		*p* < 0.001		*p* = 0.030	
Thickened pleural line	1 (0.4)	0.06	0 (0)	−0.001	0 (0)	−0.002	0 (0)	−0.004	0 (0)	−0.003	0 (0)	−0.002	0 (0)	−0.002
	*p* = 0.001		*p* = 0.972		*p* = 0.919		*p* = 0.802		*p* = 0.847		*p* = 0.917		*p* = 0.909	
Single B-line artifacts	103 (38.4)	**0.51**	2 (50.0)	0.08	6 (18.8)	0.07	68 (36.6)	0.40	33 (29.2)	0.24	7 (20.6)	0.08	16 (39.0)	0.20
	*p* < 0.001		*p* < 0.001		*p* < 0.001		*p* < 0.001		*p* < 0.001		*p* < 0.001		*p* < 0.001	
Multiple B-line artifacts	79 (29.6)	**0.50**	2 (50.0)	0.10	14 (43.8)	0.25	62 (33.3)	0.47	33 (29.2)	0.31	9 (26.5)	0.15	18 (43.9)	0.29
	*p* < 0.001		*p* < 0.001		*p* < 0.001		*p* < 0.001		*p* < 0.001		*p* < 0.001		*p* < 0.001	
B-line artifacts—white lung	14 (5.2)	0.21	0 (0)	−0.002;	11 (34.4)	0.48	15 (8.1)	0.27	7 (6.3)	0.15	15 (44.1)	**0.64**	0 (0)	−0.008
	*p* < 0.001		*p* = 0.890		*p* < 0.001		*p* < 0.001		*p* < 0.001		*p* < 0.001		*p* < 0.001	
AM artifacts	1 (0.4)	0.06	0 (0)	−0.001;	0 (0)	−0.002	1 (0.5)	0.07	0 (0)	−0.003	1 (2.9)	0.17	0 (0)	−0.002
	*p* = 0.001		*p* = 0.972		*p* = 0.919		*p* < 0.001		*p* = 0.847		*p* < 0.001		*p* = 0.909	
Consolidations < 5 mm	31 (11.6)	0.30	0 (0)	−0.004;	13 (40.6)	0.37	30 (16.2)	0.35	9 (8.0)	0.12	18 (54.5)	**0.51**	4 (9.8)	0.09
	*p* < 0.001		*p* = 0.830		*p* < 0.001		*p* < 0.001		*p* < 0.001		*p* < 0.001		*p* < 0.001	
Consolidations > 5 mm	5 (1.9)	0.13	0 (0)	−0.001;	1 (3.1)	0.07	5 (2.7)	0.16	4 (3.5)	0.16	1 (3.0)	0.07	0 (0)	−0.005
	*p* < 0.001		*p* = 0.936		*p* < 0.001		*p* < 0.001		*p* < 0.001		*p* < 0.001		*p* = 0.797	

**Table 6 jcm-10-04114-t006:** Analysis of the frequency of findings detected in LUS.

	ILD in LUS					
	No		Yes			
	*n*	%	*n*	%	*p*	RR
Pleural line—irregular	7	4.7	27	96.4	<0.001	4.82
Pleural line—coarse	2	1.4	27	96.4	<0.001	14.40
Pleural line—fragmented	2	1.4	16	57.1	<0.001	8.32
Pleural line—blurred	0	0	7	25.0	<0.001	-
Pleural line—thickened	0	0	1	3.6	0.159	-
B-line artifacts—single	2	1.4	26	92.9	<0.001	13.81
B-line artifacts—multiple	1	0.7	16	57.1	<0.001	15.72
B-line artifacts—white lung	0	0	4	14.3	0.001	-
Am artifacts	0	0	1	3.6	0.159	-
Consolidations < 5 mm	0	0	7	25.0	<0.001	-
Consolidations > 5 mm	0	0	2	7.1	0.025	-

**Table 7 jcm-10-04114-t007:** Analysis of the frequency *n* (%) of abnormalities depending on localization. The table includes numbers of detected cases; % of the sample is given in brackets.

	R-Anterior-Superior	L-Anterior-Superior	R-Anterior-Inferior	L-Anterior Inferior	R-Lateral Superior	L-Lateral Superior	R-Lateral Inferior	L-Lateral Inferior	R-Posterior-Superior K	L-Posterior Superior	R Posterior Middle-K	L-Posterior Middle-K	R-Posterior Inferior-K	L-Posterior Inferior-K	R Posterior-Middle-P	L-Posterior Middle-P	R-Posterior Inferior P	L-Posterior Inferior P
Pleural line—irregular	8 (4.5)	8 (4.5)	22 (12.5)	2 (12.5)	5 (2.8)	6 (3.4)	25 (14.2)	24 (13.6)	10 (5.7)	8 (4.5)	20 (11.4)	20 (11.4)	32 (18.2)	32 (18.2)	19 (10.8)	18 (10.2)	31 (17.6)	30 (17.0)
Pleural line—coarse	4 (2.3)	3 (1.7)	14 (8.0)	12 (6.9)	3 (1.7)	4 (2.3)	20 (11.4)	18 (10.2)	6 (3.4)	3 (1.7)	9 (5.1)	11 (6.3)	29 (16.5)	29 (16.5)	7 (4.0)	8 (4.5)	26 (14.8)	25 (14.2)
Pleural line—fragmented	2 (1.1)	1 (0.6)	6 (3.4)	6 (3.4)	1 (0.6)	3 (1.7)	9 (5.1)	10 (5.7)	1 (0.6)	1 (0.6)	5 (2.8)	9 (5.1)	13 (7.4)	13 (7.4)	3 (1.7)	2 (1.1)	15 (8.5)	13 (7.4)
Pleural line—blurred	0	0	3 (1.7)	2 (1.1)	1 (0.6)	1 (0.6)	3 (1.7)	3 (1.7)	0	0	0	0	6 (3.4)	7 (4.0)	1 (0.6)	0	5 (2.8)	6 (3.4)
Pleural line—thickened	0	1 (0.6)	0	0	0	0	0	0	0	0	0	0	0	0	0	0	0	0
B-lines—single	3 (1.7)	2 (1.1)	10 (5.7)	10 (5.7)	1 (0.6)	2 (1.1)	13 (7.4)	11 (6.3)	5 (2.8)	3 (1.7)	5 (2.8)	6 (3.4)	15 (8.5)	14 (8.0)	3 (1.7)	5 (2.8)	15 (8.5)	13 (7.4)
B-lines-multiple	1 (0.6)	2 (1.1)	4 (2.3)	4 (2.3)	3 (1.7)	3 (1.7)	6 (3.4)	8 (4.5)	1 (0.6)	1 (0.6)	4 (2.3)	5 (2.8)	10 (5.7)	11 (6.3)	3 (1.7)	3 (1.7)	9 (5.1)	9 (5.1)
B-lines—white lung	0	0	1 (0.6)	1 (0.6)	0	0	1 (0.6)	1 (0.6)	0	0	0	1 (0.6)	2 (1.1)	2 (1.1)	0	0	3 (1.7)	4 (2.3)
Am-lines	0	0	0	0	0	0	0	0	0	0	0	0	0	0	0	0	0	1 (0.6)
Consolidations < 5 mm	0	0	3 (1.7)	3 (1.7)	0	1 (0.6)	2 (1.4)	3 (1.7)	0	0	2 (1.1)	3 (1.7)	4 (2.3)	5 (2.8)	2 (1.1)	1 (0.6)	4 (2.3)	4 (2.3)
Consolidations > 5 mm	0	0	0	0	0	0	0	0	0	0	0	0	2 (1.1)	1 (0.6)	0	0	1 (0.6)	1 (0.6)

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
