# Peer review of "Lung Ultrasound in the Screening of Pulmonary Interstitial Involvement Secondary to Systemic Connective Tissue Disease: A Prospective Pilot Study Involving 180 Patients"

_jcm, 2021, doi:10.3390/jcm10184114_

Round 1
Reviewer 1 Report
The manuscript has interesting inputs but needs better clarifications and details regarding the study methodology applied and the way results are presented.
Methods:
1) The authors declare that 4/180 patients were initially excluded because of LV failure, mitral or aortic stenosis. Later, when describing the 176 patients enrolled, there are 3 LV insufficiency, 4 mitral and 3 aortic valve disease. Those patients should be excluded and the analysis repeated without them.
2) many different LUS parameters were assessed, really many. What is the feasibility of this assessment? How long is that, on each patient? What is the definition of "single", "multiple" and "white lung" B-lines?
3) HRCT can be "easily" clustered into positive or negative, according to ILD presence of absence. I am not so confident that this can be the cases of LUS. Which of the 11 parameters make the patient LUS positive, or how many of them, or which combination make a patient as "ILD on LUS" positive? This should be extensively clarified in the methods section.
4) I am not sure whether "Spearman correlation" is the appropriate method to test the relationship between two dichotomic variables (as in table 5). I would expect Chi-Square or regression analysis.
5) the novelty of LUS as a screening test for ILD is not "so novel", please see Barskova T, Gargani L et al, ARD 2013.
Author Response
Thank you for your constructive comments. Below we put the answers and give the number of the line in the text where the changes were made.
Methods:
- The authors declare that 4/180 patients were initially excluded because of LV failure, mitral or aortic stenosis. Later, when describing the 176 patients enrolled, there are 3 LV insufficiency, 4 mitral and 3 aortic valve disease. Those patients should be excluded and the analysis repeated without them.
In the cardiological assessment, the reported defects were not clinically significant, therefore these patients were included for further statistical analysis. We have completed the description of the study group (line numbers 49-51).
- many different LUS parameters were assessed, really many. What is the feasibility of this assessment? How long is that, on each patient?
The lung ultrasound test time was on average 5-7 minutes. The description of the observed pulmonary changes took about 3 minutes on average. The repeatability of the test was ensured by the established test protocol. (135-136 line)
- What is the definition of "single", "multiple" and "white lung" B-lines?
Single B lines – less than 3 in one intercostal space
Multiple B lines – more than 3 in one intercostal space
White lung – numerous B lines, which merging in a one wide vertical artifact, fill up entire intercostal space
This information had been included in the description of the table 3 (line 140-142).
- HRCT can be "easily" clustered into positive or negative, according to ILD presence of absence. I am not so confident that this can be the cases of LUS. Which of the 11 parameters make the patient LUS positive, or how many of them, or which combination make a patient as "ILD on LUS" positive? This should be extensively clarified in the methods section.
The analysis showed that a positive ultrasound result was recorded in the majority of cases when the following symptoms were present: irregular pleural line, coarse, fragmented pleural line and single and multiple B artefacts. The relative risk of interstitial disease ranged from 4.82 for irregular pleural line to 15.72 for multiple B artifacts. (Line 195-196
As these data represent additional findings of our study, we have included them in the results section rather than in the methods section. Many thanks for this reviewer's comment, as it brought to our attention this important finding of the study
4) I am not sure whether "Spearman correlation" is the appropriate method to test the relationship between two dichotomic variables (as in table 5). I would expect Chi-Square or regression analysis.
Dear Editor and Reviewer, we do not present the Spearman correlation in Table 5.
In table 5, the strength of the relationship is defined by phi and p, which are for Fisher exact / chi2 test (depending on sample size) - exactly as recommended by the reviewer. The description of the table also indicates this. (Line 192-193)
5) the novelty of LUS as a screening test for ILD is not "so novel", please see Barskova T, Gargani L et al, ARD 2013.
We corrected this sentence and added some literature (line 218)

Reviewer 2 Report
Overall I thought this was a nicely done paper which evaluated the diagnostic role of LUL for the detection of ILD. The strengths of this paper include prospective design, blinding for all 3 imaging modalities, and large group of patients with various CTD. I have a few suggestions.
-More detail would be helpful as to when in the patient's disease course they were enrolled in this study. For example, were these new CTD patients?
-Please expand demographics to include smoking status
-Would include sensitivity, specificity, PPV, and NPV in texts of results section - The PPV and NPV I think are not correct as they are currently written
-The discussion section could be made stronger - rather than summarizing the main results would place in context of prior results using LUL for detection ILD, how do the sensitivity / specificity compare, the limitations sections should expanded
-Table 1 - some of the frequencies have a comma instead of a decimal place
-Figure 2 and 3 are graphic representations of table 2 and 3 - would delete these figures
-There were a few English language errors that can be improved for clarity
Author Response
Thank you for your constructive comments. Below we put the answers and give the number of the line in the text where the changes were made.
-More detail would be helpful as to when in the patient's disease course they were enrolled in this study. For example, were these new CTD patients?
All patients included in the study had a diagnosis of systemic disease established at least 2 years previously. All of them met the classification criteria of the particular disease and had a stable form of the disease. There were no newly recognized patients in our study group. (line 116)
-Please expand demographics to include smoking status
As there were only a few people (7 patients) in the study group who were active smokers, this did not affect the results of the study. Therefore, these data were not originally included in the demographic details. This information has now been included in the descriptive data of the group. Unfortunately, we have no data on past smoking. (line 122-123)
-Would include sensitivity, specificity, PPV, and NPV in texts of results section - The PPV and NPV I think are not correct as they are currently written
Thank You very much for Your attention, we put the results into the text and correct NNP and PPV (line 166-171;line 231-233; and in the abstract)
-The discussion section could be made stronger - rather than summarizing the main results would place in context of prior results using LUL for detection ILD, how do the sensitivity / specificity compare, the limitations sections should expanded
Thank you for paying attention to the structure of the discussion. We made the appropriate changes (please find line numbers in the text) (line 264-273 and line 275-282)
-Table 1 - some of the frequencies have a comma instead of a decimal place
Is was corrected
-Figure 2 and 3 are graphic representations of table 2 and 3 - would delete these figures
The figures have been removed.
-There were a few English language errors that can be improved for clarity
The manuscript has been verified by a person fluent in English

Round 2
Reviewer 1 Report
Thanks for addressing my comments, I think the overall quality and clarity of the manuscript has improved. Still some small points are opened:
- please define "positive LUS" and "positive HRCT" in the methods.
- having "a single B line" in 1/18 fields makes the patient positive LUS? If I have a single B line in one space, then a multiple in another space, how is this patient labelled, as positive for both alterations?
- it is quite hard to believe that a "serious valvulopathy" might be "non clinically relevant". Similarly, you exclude patients for LVEF impairment and then include those who have the very same defect in the Echo done for the study. For the same reason, otherwise, you may have not excluded the initial 4 patients. I would definitely suggest to remove these patients and repeat the analysis, otherwise is it quite a meaningful methodological bias.
- discussion, lines 221-222, the method requires "validation".
Author Response
Dear Reviewer, thank you very much for the honest review and very helpful and accurate comments. Below, our manuscript correction and replies:
- “please define "positive LUS" and "positive HRCT" in the methods”,
having "a single B line" in 1/18 fields makes the patient positive LUS? If I have a single B line in one space, then a multiple in another space, how is this patient labelled, as positive for both alterations? “
Answer:
Please find our explanation in the line 90-93 for “positive HRCT” and in the line 66-69 for “positive LUS”.
- “it is quite hard to believe that a "serious valvulopathy" might be "non clinically relevant". Similarly, you exclude patients for LVEF impairment and then include those who have the very same defect in the Echo done for the study. For the same reason, otherwise, you may have not excluded the initial 4 patients. I would definitely suggest to remove these patients and repeat the analysis, otherwise is it quite a meaningful methodological bias.”
Answer:
We have carefully analysed all the data and found the place that caused the misunderstanding. We are very sorry for this misunderstanding and we appreciate the reviewer's inquisitiveness. Our haste resulted in a serious oversight.
180 patients were subjected to the study, but the manuscript contains the results of the statistical analysis of 176 patients, as 4 patients were excluded from the ultrasound and HRCT analysis due to severe disorders in ECHO – there were only 4 patients excluded because the abnormalities found in the echo overlapped. Now we see that the data we gave may have given the impression that a total of 11 people should be excluded. Of course, we have corrected the manuscript to avoid these inaccuracies.
In next lines we present results where were:
“…….Left ventricular failure (2 patients with LVEF < 50%, and 1 person with grade III left ventricular diastolic dysfunction and 1 person with LVEF < 50% and diastolic dysfunction)…”
“….The analysis revealed that 12% (n = 22) of patients demonstrated the IVC width expiration/inspiration < 50%, and 6.3% (n = 11) IVC > 21 mm. LVEF < 50% occurred in 3 patients and in the same number of patients right ventricular (RV) systolic function < 9.5cm/s and a serious defect of the aortic valve was detected. In 3.4% (n = 6) of patients RV systolic function TAPSE < 17mm was revealed, and in 2.3% (n = 4) a serious defect of the mitral valve. …”
Explanation is in the line 131-136:
The lesions in the echo of the heart overlapped in individual patients, for example: among patients with heart failure: in 3 patients it was found serious mitral valve defect (MR ⩾moderate or stenosis) and in one patient was also found a severe defect in the aortic and mitral valve.
“discussion, lines 221-222, the method requires "validation"”
Answer: We add the comments in the line 220.
